

# Applying the S-O-R model to explore impulsive buying behavior driven by influencers on social commerce websites

Tanaporn Hongsuchon[1], Shih-Chih Chen[2] and Asif Khan[3]

[1] Chulalongkorn Business School, Chulalongkorn University, Bangkok, Thailand
[2] Department of Information Management, National Kaohsiung University of Science and Technology, Kaohsiung, Taiwan
[3] College of Business, Southern Taiwan University of Science and Technology, Tainan, Taiwan

## ABSTRACT

In recent years, influencer marketing has gained increasing popularity, with many influencers embedding product information into their content (*e.g.*, videos and articles). When fans encounter these messages, they may make unplanned purchases, resulting in impulse buying behavior, a long-standing issue in marketing research. This study aims to explore the factors that lead to such behavior.

Using the Stimulus–Organism–Response (S-O-R) model as a framework, the study investigates how interactions between individuals and influencer content (Stimuli) trigger psychological changes in consumers, namely positive affect, flow state, and emotional attachment (Organism), which in turn lead to impulse buying behavior (Response). The study surveyed fans who had previously purchased products recommended by influencers, collecting 404 valid responses.

The findings reveal that: (1) Consumers' psychological changes (positive affect, flow state, and emotional attachment) significantly and positively influence impulse buying behavior. (2) Scarcity, discounted price, review quality, and observational learning also have significant positive effects on impulse buying. (3) Social presence and sense of belonging significantly enhance flow state. (4) Entertainment and informativeness significantly enhance emotional attachment.

## INTRODUCTION

Today, people may not be shopping in brick-and-mortar stores as much as before. With the advancement of technology, people often use the Internet and mobile phones to shop since there are no time and area restrictions for doing so. On the other hand, some people need to refer to other people's opinions in order to make decisions when shopping, especially for online stores. However, consumers do not always get reviews on e-commerce platforms, such as reviews of buying experiences. This problem is solved by social commerce (S-commerce), which combines e-commerce and community media (*Liu et al., 2021*). Such websites provide a platform for consumers to exchange information to help them make shopping decisions. The widespread implementation of S-commerce technologies has been instrumental in driving the global S-commerce sector to achieve a

Corresponding author
Asif Khan, khan_asif04@yahoo.com

valuation of USD 727.63 billion in 2022, with market analysts anticipating continued growth at an annual compound rate of 30.8% over the 2023–2030 period (*Moghddam et al., 2024*). S-commerce platforms fundamentally diverge from traditional electronic commerce models by emphasizing dynamic user engagement features, community-driven participation frameworks, and multimedia content elements that can trigger impulsive consumer purchasing decisions (*Ibrahim, Ariyanti & Iskamto, 2025*).

Impulse buying constitutes an instantaneous, compelling, and hedonistically complex consumer behavior pattern distinguished by limited deliberative assessment of available information and inadequate evaluation of purchasing alternatives (*Zafar et al., 2021b*). These consumption choices emerge from emotion-driven rather than logical decision-making processes. Digital S-commerce platforms intensify traditional tendencies toward unplanned purchases by leveraging real-time interactive communications, compelling user-generated multimedia content, and community-based promotional tactics (*Yang et al., 2021*).

Anyone can share their life stories and works on S-commerce websites. Some people's works are loved by others and accumulate many followers; these people are called influencers (*Ki et al., 2020*). Influencer marketing refers to companies placing the products or information they want to promote in posts or videos released by influencers on social media (*Lou & Yuan, 2019*). This placement marketing is usually called advertorial. Influencer marketing is valued because it influences fans and followers, which is much more influential than traditional marketing.

Although impulse buying can bring business opportunities, it also negatively impacts consumers and enterprises. For example, consumers feel regret or write negative reviews on the Internet (*Abdelsalam et al., 2020*). Therefore, when companies on social business websites understand consumer impulse purchase behaviors and trigger factors, they can quickly adjust their business strategies and respond to market demand promptly. Research has explored the impact of specific information communication techniques (*e.g.*, posting) on impulse buying behavior on S-commerce websites. For instance, *Leong, Jaafar & Ainin (2018)* undertook a quantitative analysis examining how Facebook commerce navigation behaviors and engagement frequency affected consumers' buying impulses and resulting unplanned purchase activities. Furthermore, *Ming et al. (2021)* endeavored to clarify the pathways by which perceived social presence in live-streaming contexts shapes consumer confidence and psychological flow experiences, consequently triggering immediate buying decisions. In addition, *Zafar et al. (2021a)* investigated the interconnections among celebrity influencer content genuineness, emotional tone characteristics, observational learning mechanisms, and consumers' inclination toward spontaneous purchasing conduct.

This research fills a significant theoretical and empirical void in S-commerce studies. It investigates distinct psychological pathways whereby influencer-driven stimuli precipitate impulsive buying behavior. This phenomenon differs markedly from traditional e-commerce environments. Current e-commerce research has thoroughly examined impulse buying through isolated factors like price promotions or website design (*Ruangkanjanases et al., 2024*). However, these studies cannot capture the multifaceted

stimulus environment of S-commerce platforms. In these platforms, commercial, social, and para-social elements function simultaneously (*Ibrahim, Ariyanti & Iskamto, 2025*). The contribution of applying the stimulus-organism-response (S-O-R) model here is its theoretical integration of three interconnected stimulus pathways. These include commercial stimuli (discounted prices, scarcity), social interaction stimuli (social presence, sense of belonging), and influencer-specific stimuli (entertainment, informativeness). Together, they generate compound psychological effects that are absent in conventional online retail environments.

This research addresses a crucial gap in comprehending how para-social relationships with influencers establish distinctive pathways to impulse buying through emotional attachment (*Pang & Zhang, 2024*). Traditional e-commerce models fail to explain this mechanism adequately. Previous studies have examined S-commerce features separately. This investigation demonstrates how multiple stimulus types function synergistically to trigger various psychological states (positive affect (*Tu, Huang & Tu, 2025*), flow state (*Husada, Prawiyadi & Andreani, 2024*), emotional attachment (*Sánchez-Fernández & Jiménez-Castillo, 2021*)) concurrently. This creates stronger behavioral responses than individual stimuli working alone. This multi-pathway framework enhances S-O-R theory by showing that S-commerce environments demand complex, parallel mediation models instead of simple stimulus-response connections.

The theoretical contribution goes beyond validating established relationships to uncovering the psychological framework underlying S-commerce effectiveness. The study identifies how observational learning (*Lu et al., 2021*) and social presence (*Mahmood, Jusoh & Nor, 2024*) generate flow states that circumvent rational decision-making. Meanwhile, entertainment (*Han, 2025*) and informativeness (*Choi & Kim, 2023*) develop emotional attachments that motivate purchase behavior. This investigation offers a thorough framework for understanding consumer psychology within digitally mediated social shopping environments. These results advance emerging S-commerce theory by defining theoretical distinctions between traditional e-commerce and socially-embedded commerce. This provides scholars with a solid foundation for future investigations into psychological mechanisms that influence consumer behavior in increasingly social digital marketplaces.

## LITERATURE REVIEW AND HYPOTHESIS DEVELOPMENT

### Social commerce (S-commerce)

S-commerce is a business type that combines social media and e-commerce. Consumers use social media and content others post to help themselves make decisions and purchase goods online (*Moghddam et al., 2024*). Nevertheless, the conceptual separation between s-commerce and traditional e-commerce goes beyond technological integration alone. Traditional e-commerce emphasizes transactional efficiency and product-centered interactions primarily. S-commerce fundamentally reshapes the buying process by integrating social interactions, peer influence, and community-driven decision-making within commercial contexts (*Ibrahim, Ariyanti & Iskamto, 2025*). This evolution produces what academics describe as "social shopping." Here, purchasing decisions receive influence

from social networks, user-generated content, and collective intelligence instead of solely from product attributes and price comparisons (*Moghddam et al., 2024*). Social influence theory (*Hazari, Talpade & Brown, 2024*) provides the theoretical basis for s-commerce. This framework recognizes three influence categories: compliance (behavioral change due to rewards/punishments), identification (behavioral change to maintain relationships), and internalization (behavioral change due to value alignment). Traditional e-commerce operates predominantly through compliance mechanisms like price incentives and promotions. S-commerce employs all three influence categories *via* social elements such as peer recommendations, influencer endorsements, and community participation (*Zhu, Tan & Panwar, 2024*). This complex influence framework creates distinctive pathways to impulse buying that are unavailable in standard e-commerce settings.

These platforms based on social media to expand into S-commerce are called S-commerce websites. On the other hand, people increasingly rely on online information to understand products and services when shopping online. However, information such as likes, reviews, and ratings can no longer meet the needs of consumers (*Ho & Rajadurai, 2020*). Live streaming has been developed and introduced into e-commerce to compensate for the lack of face-to-face and real-time interaction online. Through live streaming, consumers can chat with sellers in real-time, thus stimulating consumers' interest and purchase intention (*Clement Addo et al., 2021*; *Ho & Rajadurai, 2020*).

S-commerce websites such as YouTube, Facebook, Line, and Instagram, which are widely used today, have post and live-streaming functions. These platforms have a significant impact on consumer purchase decisions. Many researchers have discussed online impulse buying behavior. For instance, *Xu, Zhang & Zhao (2020)* executed their analysis through the S-O-R theoretical framework integrated with dual systems theory to explore how social interaction mechanisms and self-regulatory processes influence consumers' impulsive purchasing behaviors. This study argues that impulse buying behavior on S-commerce websites should also be explored.

### Influencer marketing

Influencer marketing refers to a marketing strategy that uses the influence of key opinion leaders to drive consumers to make purchasing decisions (*Choi, Wu & Lee, 2025*). Marketing and media are interdependent. Marketing companies use media to push advertisements to attract target audiences. In addition, celebrity endorsement is one of the popular types of marketing (*Lou & Yuan, 2019*). However, with the changing times, social media has become people's main information source. This has led to the rise of celebrities, such as influencers, defined as those with a large following on social media (*Haenlein et al., 2020*). Research indicates that netizens post their creations (*e.g.*, text or video) on social media and have large followings, while consumers believe they are more trustworthy than traditional celebrities (*Jin, Muqaddam & Ryu, 2019*). For instance, *Al-Shehri (2021)* examined how social media influencers impact consumer purchasing intentions. The study also explored whether gender differences among consumers or influencers influenced these decision-making processes. Another example is *Lou & Yuan*'s *(2019)* study, which constructed a comprehensive theoretical framework called the social media influencer

value model. This framework explains the roles of advertising value perceptions and source credibility factors. On the other hand, social media can communicate faster, have a broader target audience, get more feedback, and cost less than traditional media (*Lou & Yuan, 2019*). Advertisements created and published by influencers can also gain consumers' trust more than traditional advertisements. Therefore, influencer marketing is considered more cost-effective and effective (*Lim et al., 2017*; *Lou & Yuan, 2019*).

Some researchers have pointed out that influencers regularly publish works on social media, which makes fans highly favor the brands or products they introduce, affecting online purchase intention and impulse purchase behavior. For instance, research by *Trivedi (2021)* compared the relative effectiveness between attractive celebrity influencers and expert influencers in promoting online consumer-brand engagement behaviors. This subsequently resulted in spontaneous online purchasing decisions. Another example is research conducted by *Lim et al. (2017)* that investigated the effectiveness of social media influencers by analyzing essential factors. These factors included source credibility perceptions, source attractiveness attributes, product congruence alignment, and meaning transfer processes. According to the previous literature, this study focuses on the factors that lead to impulsive buying behavior on S-commerce websites due to influencers.

## Stimulus-organism-response model (S-O-R model)

The S-O-R model was first proposed by *Woodworth (1929)*. It has been applied to environmental psychology to reveal that environmental stimulus (S) affects individuals' cognition and emotion (O), which in turn affects individual behavioral responses (R) (*Mehrabian & Russell, 1974*). The S-O-R model demonstrates a notable theoretical enhancement compared to traditional behaviorist stimulus-response models. It distinctly integrates the mediating influence of internal psychological processes. The S-O-R model merges these approaches by identifying the organism component as the vital mediating mechanism. This mechanism facilitates the transformation of external stimuli into behavioral responses.

In recent years, *Yang et al. (2021)* have used S-O-R models to explore the factors affecting impulse buying behavior. *Yang et al. (2021)* empirically assessed scientific hypotheses regarding whether Internet utilization elevates life satisfaction levels among elderly populations. The investigation examined whether differences appear in the impact of Internet usage on elderly well-being outcomes. Moreover, it determined through which mechanisms Internet engagement affects elderly individuals' life satisfaction measures. Moreover, *Chan, Cheung & Lee (2017)* argue that online environmental stimuli are the key factors that affect impulse buying. Therefore, the present research uses the S-O-R model as a framework to discuss impulse buying behavior. Next, the three concepts of stimulus, organism, and response are discussed further.

### Stimulus

The theoretical consistency of this stimulus framework stems from its acknowledgment that S-commerce environments generate interconnected stimulus networks instead of isolated environmental cues. This integrated stimulus conceptualization corresponds with

the S-O-R model's foundation that environmental complexity necessitates comprehensive stimulus consideration for predicting accurate psychological and behavioral responses (*Mehrabian & Russell, 1974*). The constructs incorporated in the stimulus framework receive the explanation below.

Scholars believe that the content and environment on the website and users' perceptions influence purchase behavior intentions (*Chan, Cheung & Lee, 2017*). For example, companies often use promotional campaigns to stimulate consumers' purchase intentions (*Abolghasemi et al., 2020*). There are many kinds of it. Among them, price discounts make consumers feel different from their past experiences and stimulate, producing a response (*Sheehan et al., 2019*). The theoretical foundation of price discount effects originates from prospect theory (*Han et al., 2025*), which proposes that consumers assess gains and losses concerning a reference point. Price discounts generate a perception of gain, activating positive emotional responses that can supersede rational decision-making processes (*Zeng et al., 2025*). Within S-commerce contexts, this effect becomes magnified through social proof mechanisms. Discounted prices are frequently presented alongside social engagement metrics (likes, shares), producing compound stimulus effects (*Delao & Myers, 2021*).

In addition, scarcity (*e.g.*, limited time, limited quantity) makes consumers feel a lack of resources and threats, increasing the demand for goods (*Chen & Yao, 2018*). Scarcity appeals function through psychological reactance theory (*Rosenberg & Siegel, 2025*), whereby perceived limitations on the freedom to acquire a product enhance its desirability. The scarcity principle proves especially effective in S-commerce as it merges temporal urgency with social validation. When consumers witness others interacting with scarce products, the fear of missing out (FOMO) becomes more intense (*Yu et al., 2024*); this, combined mechanism of scarcity and social observation, establishes a potent stimulus environment capable of triggering impulsive responses (*Shi, Li & Chumnumpan, 2020*). Past research has indicated that discounted prices (*Büyükdağ, Soysal & Kitapci, 2020*; *Sheehan et al., 2019*) and scarcity (*Yu et al., 2024*) significantly affect consumers' purchase intentions.

On the other hand, many people rely heavily on social network sites (SNSs), read others' reviews (*Zhao, Stylianou & Zheng, 2018*), and observe and learn from others' behavior (*Zafar et al., 2021b*) before shopping. Review quality, operationally characterized as user-generated evaluations' perceived credibility, relevance, and comprehensiveness, fulfills multiple roles in S-commerce environments. Reviews function beyond providing product information by acting as social proof mechanisms that minimize uncertainty and confirm purchase decisions (*Qian et al., 2021*). Within S-commerce contexts, review quality includes textual content, multimedia components (photos, videos), and social engagement metrics (likes, helpfulness ratings) that collectively indicate community approval and product appeal.

Review quality is a credibility indicator that decreases information asymmetry between sellers and buyers (*Zafar et al., 2021b*). In S-commerce environments, reviews serve as informational tools and social indicators that demonstrate community approval and social acceptance (*Xu, 2020*).

Additionally, observational learning, based on Bandura's social learning theory (*Zhou et al., 2024*), constitutes a cognitive process through which consumers gain knowledge, attitudes, and behaviors by observing others' actions and outcomes. In S-commerce environments, observational learning occurs through multiple channels: (1) direct observation of influencer product usage and consumption experiences, (2) indirect learning through peer reviews and testimonials, and (3) social modeling through community interactions and shared experiences (*Lu et al., 2021*). Review quality (*Li, Wu & Mai, 2019*) and observational learning (*Zafar et al., 2021b*) influence purchase decisions.

Moreover, consumers' perceptions influence their purchase intentions, especially on S-commerce websites. The psychological dimension of stimulus perception gains particular importance in S-commerce due to the inherently social character of these platforms. Two essential perceptual constructs emerge as primary drivers of consumer responses: social presence and sense of belonging. Social presence, theoretically founded on media richness theory (*Ngo et al., 2025*), is operationally characterized as the extent to which consumers perceive influencers and other community members as psychologically present and available during their S-commerce experience. In S-commerce contexts, social presence is enabled through technological capabilities such as live streaming, real-time chat, instant messaging, and interactive features replicating face-to-face interactions (*Mahmood, Jusoh & Nor, 2024*).

Based on belongingness theory (*Gao, Liu & Li, 2017*), a sense of belonging constitutes a fundamental human need to feel connected, accepted, and valued by a social group. Operationally, this construct is characterized as consumers' perceived integration into the S-commerce community, marked by social acceptance, shared identity, emotional connection, and participatory engagement (*Pang & Zhang, 2024*). In S-commerce environments, a sense of belonging is developed through user-generated content participation, social sharing activities, community discussions, and collaborative consumption experiences (*Cho & Son, 2019*). With the advancement of technology, S-commerce websites have more and more functions. Through these features, consumers perceive that they can become closer to influencers, thereby generating a social presence (*Wang et al., 2021*) and increasing interactions with others and also perceive a sense of belonging (*Li & Guo, 2021*).

In addition, S-commerce sites can be pleasant and relaxing and provide helpful information (*Chen & Lin, 2018*). Some researchers have indicated that the entertainment and informativeness of S-commerce websites significantly affect purchase intention (*Ki et al., 2020*). The entertainment value in S-commerce originates from the uses and gratifications theory (*Nguyen & Nguyen, 2024*), which describes how individuals actively pursue media content to fulfill specific needs. Entertainment gratification encompasses enjoyment, escapism, and aesthetic pleasure from influencer content (*Han, 2025*).

Additionally, informativeness, based on information processing theory (*Payne, 2024*), denotes the degree to which content delivers valuable, relevant, and timely information for decision-making. In S-commerce, informativeness extends beyond product specifications to encompass contextual information such as usage scenarios, styling tips, and

comparative evaluations provided by influencers (*Choi & Kim, 2023*). The dual-process theory indicates that informative content can affect behavior through both central (systematic processing) and peripheral (heuristic processing) pathways, depending on consumer involvement and motivation (*Xu, Zhang & Zhao, 2020*). Based on past research, this study argues that promotional campaigns (including discounted prices and scarcity), review quality, observational learning, social presence, sense of belonging, entertainment, and informativeness stimulate consumers' purchase intentions, so they are considered "stimulus".

### Organism

The organism is how individuals mentally translate a stimulus into a message (*Mehrabian & Russell, 1974*). It is an emotional response generated when consumers interact with a stimulus (*Chen & Yao, 2018*). Previous research has suggested that positive affect (*Xu, Zhang & Zhao, 2020*), flow state (*Ming et al., 2021*), and emotional attachment (*Ki et al., 2020*) are organisms and influence purchase behavioral intention. Therefore, this research model used these three factors as the organism. Next, positive affect, flow state, emotional attachment, and the relationship between stimulus and organism are further described.

Positive affect refers to an individual's subjective experience of happiness and excitement (*Bandyopadhyay et al., 2021*).

Whether researched in brick-and-mortar or online stores, positive emotions are associated with impulse buying behavior. For instance, *Chan, Cheung & Lee (2017)* conducted a thorough systematic review of research investigating online impulsive purchasing behaviors, using the S-O-R theoretical framework to identify and classify the factors influencing online spontaneous buying decisions. Another instance can be a study by (*Chen & Yao, 2018*) that employed mobile auction platforms as their research setting to examine how situational variables affect impulsive purchasing behaviors, incorporating an integrated S-O-R model that included impulsivity personality traits along with other external factors.

On the other hand, promotional campaigns (discounted price and scarcity) significantly impacts positive affect. Scholars point out that discounted prices make consumers feel pleasant. Positive affect is higher when consumers feel more strongly about discounted prices (*Bandyopadhyay et al., 2021*). Furthermore, the scarcity of a commodity enhances consumers' perception of its value, and a positive affect is generated (*Chen & Yao, 2018*). Additionally, many consumers watch other people's reviews online or learn from the behavior of others (*e.g.*, influencers) before making a purchase decision because these behaviors provide consumers with peace of mind. *Xu, Zhang & Zhao (2020)* argue that high-quality online reviews can lead consumers to generate positive affect. Consumers interact with others and learn from others' behavior through SNSs, generating positive affect (*Zafar et al., 2021a*). Therefore, based on previous research, this study proposes the following hypothesis.

*H1a. Discounted price positively impacts positive affect.*
*H1b. Scarcity positively impacts positive affect.*
*H2. Review quality positively impacts positive affect.*

*H3. Observational learning positively impacts positive affect.*

Flow state refers to a state of mind. When an individual is in this state, he or she is very focused on the current activity, undisturbed by external factors, and feels enjoyment, happiness, and excitement (*Chen & Lin, 2018*). In other words, when people are engaged in and immersed in an activity, they are in a flow state. Past research has indicated that people in a flow state lose their rationality and generate impulse buying behaviors. For example, *Shahpasandi, Zarei & Nikabadi (2020)* examined how hedonic browsing behaviors and flow states, operating as intrinsic motivational factors, affect Instagram users' cognitive and affective experiences and their resulting impulsive purchasing behaviors. Furthermore, *Wang et al. (2021)* investigated the impacts of live broadcast characteristics on consumers' perceptions of social presence and flow experiences, along with their influence on consumption intentions within live e-commerce environments, using survey methodology.

In addition, as mentioned earlier, social presence and a sense of belonging are both psychological states generated by users' interactions with others on social media. *Ming et al. (2021)* found that social presence positively affects the flow state. *Hsu (2020)* and *Guan et al. (2022)* argue that a sense of belonging also positively affects the flow state. Thus, this study proposes the following hypothesis.

*H4. Social presence positively impacts the flow state.*

*H5. Sense of belonging positively impacts the flow state.*

Emotional attachment is defined as the emotional connection between people and is a relationship-based structure (*Bowlby, 1977*). It is important in predicting consumer behavior (*Pang & Zhang, 2024*). *Ki et al. (2020)* stated that fans develop emotional attachments to influencers, affecting their purchasing behavior. They also found that the characteristics and creations of the influencers generated entertainment and informativeness that can affect emotional attachment. Furthermore, it was found that companies publish entertaining brand or product information through social media, and then the target audiences build emotional attachments with the companies (*Yan et al., 2024*). *Sánchez-Fernández & Jiménez-Castillo (2021)* found that when influencers provide informative works, fans develop emotional attachments to them. Thus, this study proposes the following hypothesis.

*H6. Entertainment positively impacts emotional attachment.*

*H7. Informativeness positively impacts emotional attachment.*

### Response

The response is defined as outcomes and decisions made by individuals based on cognition and emotion (*Mehrabian & Russell, 1974*). On the other hand, impulse buying behavior is an important issue in consumer behavior research. Impulse buying behavior is the actual behavior of consumers who experience environmental stimuli and undergo psychological transformation (*Chan, Cheung & Lee, 2017*). Hence, this study uses impulse buying behavior as the response in the research model. Past research has shown that positive affect (*Chen & Yao, 2018*), flow state (*Ming et al., 2021*), and emotional attachment (*Chen, Yeh & Lee, 2021*) are all related to impulse buying behaviors. As mentioned earlier, this study proposes the following hypotheses.

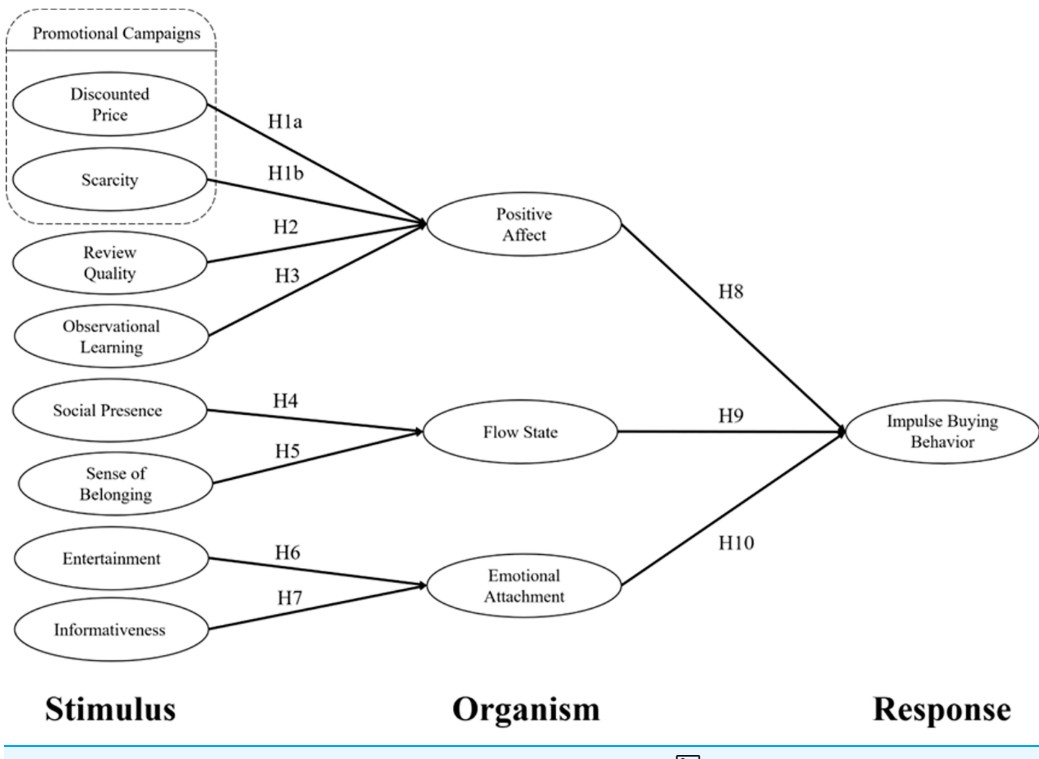

**Figure 1  Research model.**               

*H8: Positive affect positively impacts impulse buying behavior.*
*H9: Flow state positively impacts impulse buying behavior.*
*H10: Emotional attachment positively impacts impulse buying behavior.*

This research uses the S-O-R model as a framework to explore the factors that lead to the impulsive buying behavior of influencers on S-commerce websites. This research model was constructed through literature review and hypothesis development, as shown in Fig. 1.

## RESEARCH METHOD

### Research subjects and data collection

In this study, influencers are defined as creators and entertainers with over 10,000 followers and who have included advertorials in their creations or posts. This study used an online questionnaire for data collection.

Data was collected using a structured online questionnaire delivered through the Qualtrics platform for 4 weeks (March–April 2023). Participants were obtained through a multi-stage strategy combining convenience and snowball sampling techniques. Initial recruitment employed social media advertisements on Facebook, Instagram, and LINE, targeting users aged 18–45 who followed at least one influencer and had completed purchases through S-commerce platforms within the preceding 6 months. Screening questions confirmed eligibility requirements before granting survey access. The questionnaire took approximately 15–20 min to finish, with progress indicators and attention checks integrated throughout to sustain engagement and ensure data quality.

This study was reviewed by the institutional ethics committee at the National Kaohsiung University of Science and Technology, Taiwan. It was classified as exempt from formal ethical approval, as it involved minimal participant risk and solely relied on previously collected data without engaging directly with individuals. Prior to data collection, individuals were provided with comprehensive written information detailing the study's aims, procedures, and any potential risks. Written informed consent was obtained, confirming that participation was voluntary and based on understanding the research purpose and participants' rights.

Comprehensive data cleaning protocols were applied to ensure response validity. Initial screening removed 48 responses (10.7% of total submissions) according to established exclusion criteria: (1) incomplete responses containing more than 20% missing data, (2) responses finished in under 8 min (suggesting inadequate attention), (3) straight-line responses throughout multiple construct scales, (4) failed attention checks incorporated within questionnaire sections, and (5) responses from participants who failed to meet eligibility standards (no influencer following or S-commerce purchase history). Supplementary quality assessments included analyzing response patterns for outliers using Mahalanobis distance ($p < 0.001$) and detecting multivariate outliers. The final dataset contained 404 valid responses, achieving an 89.3% retention rate following data cleaning protocols, ensuring reliable statistical analysis while preserving sample representativeness.

## Research design

This study used an online questionnaire. The design of the questions for each of the constructs refers to previous research, including questions about discounted prices and scarcity were taken from *Chen & Yao (2018)*; questions about review quality, observed learning, and positive affect were taken from *Xu (2020)*; questions about social presence were taken from *Wang et al. (2021)*; questions about sense of belonging were referred to *Teo et al. (2003)*, *Lin (2008)*, and *Zolkepli & Kamarulzaman (2015)*; the questions regarding flow state were referred to *Ming et al. (2021)*, and *Wang et al. (2021)*; the questions about entertainment and informativeness were taken from *Ki et al. (2020)*; the questions about emotional attachment were referred to *Aw & Labrecque (2020)*, and *Ki et al. (2020)*; the questions about impulse purchase behavior were referred to *Chen & Yao (2018)*, *Xu (2020)*, and *Ming et al. (2021)*. In addition, this study used the Likert seven-point scale. A total of 7 means strongly agree and 1 means strongly disagree. After the questionnaire was designed, it was translated into Chinese, and experts were invited to assist in revision to improve the quality of the questionnaire. The questionnaire items of this study are indicated in the appendix section of this research.

## RESEARCH RESULTS
### Descriptive statistical analysis
#### Demographics

The results of the survey demographics are shown in Table 1. Among the 404 valid questionnaires, the majority of respondents were female, with 292 (72.2%); the age of the respondents mainly was 21~30 (62.5%); the occupation of the respondents was mostly

**Table 1 Demographic statistics (N = 404).**

| Measures | Items | Frequency | Percent (%) |
|---|---|---|---|
| Gender | Male | 112 | 27.7% |
| | Female | 292 | 72.2% |
| Age | 20 or below | 61 | 15.1% |
| | 21–30 | 252 | 62.4% |
| | 31–40 | 72 | 17.8% |
| | 41–50 | 15 | 3.7% |
| | 51–60 | 4 | 1.0% |
| Occupation | Military personnel, civil servants, or teachers | 41 | 10.1% |
| | Information technology industry | 22 | 5.4% |
| | Service industry | 80 | 19.8% |
| | Manufacturing | 46 | 11.4% |
| | Financial and insurance industry | 10 | 2.5% |
| | Student | 189 | 46.8% |
| | Freelance | 3 | 0.7% |
| | Health Care industry | 3 | 0.7% |
| | Between jobs | 5 | 1.2% |
| | Others | 5 | 1.2% |
| Monthly income (NT$) | 10,000 or below | 148 | 36.6% |
| | 10,001~30,000 | 101 | 25.0% |
| | 30,001~50,000 | 120 | 29.7% |
| | 50,001~70,000 | 26 | 6.4% |
| | 70,001~100,000 | 7 | 1.7% |
| | 100,001 or above | 2 | 0.5% |
| Frequently used social networking sites (multiple choice) | Facebook | 271 | 67.1% |
| | Instagram | 315 | 78.0% |
| | YouTube | 272 | 67.3% |
| | Weibo | 23 | 5.7% |
| | Others | 2 | 0.5% |
| Always pay attention to the type of Internet celebrity (multiple choice) | Life and entertainment | 264 | 65.3% |
| | Knowledge and education | 160 | 39.6% |
| | Cuisine and cooking | 221 | 54.7% |
| | Cosmetic and fashion | 128 | 31.7% |
| | Parent-child issues | 68 | 16.8% |
| | Illustrators | 80 | 19.8% |
| | Pets | 131 | 32.4% |
| | 3C and games | 70 | 17.3% |
| | Performing artists | 71 | 17.6% |
| | Cars and transportation | 18 | 4.5% |
| | Social issues | 59 | 14.6% |

| Table 1 (continued) | | | |
| --- | --- | --- | --- |
| Measures | Items | Frequency | Percent (%) |
| Frequency of using social networking sites (times/week) | 1–3 | 101 | 25.0% |
| | 4–6 | 85 | 21.0% |
| | 7–10 | 55 | 13.6% |
| | 11 or above | 163 | 40.3% |
| Number of Internet celebrities followed on social networking sites | 1–3 | 79 | 19.6% |
| | 4–6 | 131 | 32.4% |
| | 7–10 | 57 | 14.1% |
| | 11 or above | 137 | 33.9% |

students, with 189 (46.8%); the majority of respondents with a monthly income of less than NT$10,000, with 148 (36.6%); the most respondents follow more than 10 Influencers on S-commerce websites, with 137 (33.9%); respondents use S-commerce websites more than 10 times a week, with 163 (40.3%); the most frequently used S-commerce websites are Instagram, Facebook and YouTube; the types of influencers are life and entertainment, cuisine and cooking, and knowledge and education.

### Descriptive statistics of the sample

Table 2 shows the descriptive statistics of the questionnaire. This study used the Likert scale. The mean (4.270~5.405) is greater than the median (4), indicating that the respondents tend to agree with the questions. In addition, the standard deviation is used to understand the degree of variation in the respondents' perceptions of the questions. Table 2 shows that the standard deviations ranged from 0.903~1.327, indicating moderate agreement with the questions.

### Reliability, validity, and model fit

Next, this study examined the measurement model using confirmatory factor analysis (CFA) and analyzed the reliability, convergent validity, and discriminant validity according to *Anderson & Gerbing (1988)*. *Fornell & Larcker (1981)* proposed using composite reliability (CR), Cronbach's alpha, and factor loading to analyze questionnaire reliability. These values must be greater than 0.7; however, values greater than 0.6 are acceptable for exploratory research (*Hair et al., 2019*). Table 3 shows that CR, Cronbach's alpha, and factor loading are greater than 0.6, indicating the questionnaire is reliable.

Convergent validity is judged based on the average variance extracted (AVE), which should be greater than 0.5. Table 3 shows that the AVE ranges from 0.571 to 0.796, indicating the questionnaire has convergent validity. In addition, Fornell-Larcker and heterotrait-monotrait ratio (HTMT) were used to examine the discriminant validity. *Fornell & Larcker (1981)* proposed that the square root of AVE should be greater than the correlation coefficient between the construct and other constructs in the model. HTMT refers to the estimation of correlations between constructs. An HTMT less than 0.90 has discriminant validity (*Henseler, Ringle & Sarstedt, 2015*). Tables 4 and 5 show that the values meet the criteria, indicating that the questionnaire has good discriminant validity.

**Table 2 Descriptive statistics (N = 404).**

| Constructs | Minimum | Maximum | Mean | Standard deviation |
|---|---|---|---|---|
| Scarcity (SCA) | 1.000 | 7.000 | 4.736 | 1.166 |
| Discounted Price (DP) | 1.000 | 7.000 | 4.622 | 1.067 |
| Review Quality (RQ) | 1.000 | 7.000 | 4.353 | 1.059 |
| Observational Learning (OL) | 1.000 | 7.000 | 5.405 | 0.929 |
| Positive Affect (PA) | 1.000 | 7.000 | 4.897 | 1.134 |
| Social Presence (SP) | 1.000 | 7.000 | 4.807 | 1.067 |
| Sense of Belonging (SOB) | 1.000 | 7.000 | 4.290 | 1.206 |
| Flow State (FS) | 1.000 | 7.000 | 4.462 | 1.205 |
| Entertainment (EN) | 1.500 | 7.000 | 5.405 | 0.903 |
| Informativeness (IM) | 1.670 | 7.000 | 4.899 | 1.082 |
| Emotional Attachment (EA) | 1.400 | 7.000 | 4.812 | 0.995 |
| Impulse Buying Behavior (IBB) | 1.000 | 7.000 | 4.270 | 1.327 |

**Table 3 Reliability and validity analysis.**

| Constructs | Items | Factor loadings | VIF | Cronbach's alpha | CR | AVE |
|---|---|---|---|---|---|---|
| Scarcity (SCA) | SCA1 | 0.831 | 1.932 | 0.769 | 0.851 | 0.589 |
| | SCA2 | 0.813 | 1.902 | | | |
| | SCA3 | 0.708 | 1.657 | | | |
| | SCA4 | 0.709 | 1.652 | | | |
| Discounted Price (DP) | DP1 | 0.733 | 1.239 | 0.643 | 0.807 | 0.583 |
| | DP2 | 0.765 | 1.255 | | | |
| | DP3 | 0.792 | 1.281 | | | |
| Review Quality (RQ) | RQ1 | 0.835 | 1.933 | 0.877 | 0.915 | 0.730 |
| | RQ2 | 0.893 | 2.773 | | | |
| | RQ3 | 0.869 | 2.538 | | | |
| | RQ4 | 0.819 | 2.023 | | | |
| Observational Learning (OL) | OL1 | 0.861 | 2.137 | 0.837 | 0.891 | 0.673 |
| | OL2 | 0.873 | 2.328 | | | |
| | OL3 | 0.844 | 2.043 | | | |
| | OL4 | 0.689 | 1.495 | | | |
| Positive Affect (PA) | PA1 | 0.898 | 3.364 | 0.915 | 0.940 | 0.796 |
| | PA2 | 0.888 | 3.128 | | | |
| | PA3 | 0.896 | 3.692 | | | |
| | PA4 | 0.887 | 3.503 | | | |
| Social Presence (SP) | SP1 | 0.808 | 2.863 | 0.818 | 0.873 | 0.580 |
| | SP2 | 0.846 | 3.075 | | | |
| | SP3 | 0.736 | 2.418 | | | |
| | SP4 | 0.692 | 2.318 | | | |
| | SP5 | 0.713 | 1.461 | | | |

| Constructs | Items | Factor loadings | VIF | Cronbach's alpha | CR | AVE |
|---|---|---|---|---|---|---|
| Sense of Belonging (SOB) | SOB1 | 0.901 | 2.476 | 0.842 | 0.904 | 0.760 |
| | SOB2 | 0.862 | 2.150 | | | |
| | SOB3 | 0.851 | 1.748 | | | |
| Flow State (FS) | FS1 | 0.862 | 2.512 | 0.844 | 0.895 | 0.681 |
| | FS2 | 0.842 | 2.398 | | | |
| | FS3 | 0.777 | 2.082 | | | |
| | FS4 | 0.819 | 2.251 | | | |
| Entertainment (EN) | EN1 | 0.857 | 2.137 | 0.839 | 0.892 | 0.675 |
| | EN2 | 0.829 | 2.242 | | | |
| | EN3 | 0.840 | 2.268 | | | |
| | EN4 | 0.755 | 1.440 | | | |
| Informativeness (IM) | IM1 | 0.858 | 1.888 | 0.838 | 0.903 | 0.756 |
| | IM2 | 0.886 | 2.211 | | | |
| | IM3 | 0.864 | 1.911 | | | |
| Emotional Attachment (EA) | EA1 | 0.706 | 1.486 | 0.811 | 0.869 | 0.571 |
| | EA2 | 0.783 | 1.897 | | | |
| | EA3 | 0.817 | 2.027 | | | |
| | EA4 | 0.771 | 1.582 | | | |
| | EA5 | 0.693 | 1.434 | | | |
| Impulse Buying Behavior (IBB) | IBB1 | 0.834 | 2.596 | 0.915 | 0.935 | 0.706 |
| | IBB2 | 0.872 | 2.959 | | | |
| | IBB3 | 0.863 | 2.897 | | | |
| | IBB4 | 0.897 | 3.718 | | | |
| | IBB5 | 0.716 | 1.610 | | | |
| | IBB6 | 0.847 | 2.630 | | | |

**Notes:**
VIF, Variance Inflation Factor; CR, Composite Reliability; AVE, Average Variance Extracted.

Before examining the structural model, the research model should be examined for multicollinearity. Multicollinearity refers to whether a high correlation between constructs makes the estimation inaccurate. In this study, the variance inflation factor (VIF) was used to determine whether the model had multicollinearity. If the VIF is greater than 5, the research model has multicollinearity. Table 3 shows that the VIF values range from 1.239 to 3.718, indicating no multicollinearity.

## Structural equation modeling analysis

### Hypothesis testing

After testing the measurement model, it was confirmed that the research questionnaire met the criteria of reliability and validity. Then, the structural model was tested; the purpose was to check the relationship between the constructs, the path coefficient, and $R^2$ are used to detect. The path coefficient is the strength of the relationship between the constructs, and the larger it is, the stronger the relationship is; $R^2$ refers to the degree to which the

**Table 4 Fornell-larker discriminant validity.**

| Constructs | SCA | DP | RQ | OL | PA | SP | SOB | FS | EN | IM | EA | IBB |
|---|---|---|---|---|---|---|---|---|---|---|---|---|
| SCA | **0.767** | | | | | | | | | | | |
| DP | 0.435 | **0.764** | | | | | | | | | | |
| RQ | 0.246 | 0.373 | **0.854** | | | | | | | | | |
| OL | 0.437 | 0.409 | 0.221 | **0.820** | | | | | | | | |
| PA | 0.448 | 0.486 | 0.431 | 0.457 | **0.892** | | | | | | | |
| SP | 0.352 | 0.382 | 0.485 | 0.421 | 0.549 | **0.761** | | | | | | |
| SOB | 0.393 | 0.49 | 0.591 | 0.341 | 0.714 | 0.627 | **0.872** | | | | | |
| FS | 0.440 | 0.398 | 0.471 | 0.344 | 0.59 | 0.709 | 0.697 | **0.825** | | | | |
| EN | 0.339 | 0.334 | 0.272 | 0.547 | 0.492 | 0.512 | 0.433 | 0.551 | **0.821** | | | |
| IM | 0.382 | 0.433 | 0.501 | 0.39 | 0.543 | 0.558 | 0.603 | 0.598 | 0.595 | **0.869** | | |
| EA | 0.443 | 0.501 | 0.45 | 0.431 | 0.622 | 0.559 | 0.604 | 0.613 | 0.606 | 0.699 | **0.755** | |
| IBB | 0.500 | 0.592 | 0.414 | 0.29 | 0.577 | 0.421 | 0.59 | 0.525 | 0.284 | 0.479 | 0.573 | **0.840** |

Notes:
SCA, Scarcity, DP, Discounted Price, RQ, Review Quality, OL, Observational Learning, PA, Positive Affect, SP, Social Presence, SOB, Sense of Belonging, FS, Flow State, EN, Entertainment, IM, Informativeness, EA, Emotional Attachment, IBB, Impulse Buying Behavior; the value of the diagonal is the square root of AVE.
Bold numbers on the diagonal represent the square root of the AVE for each construct.

**Table 5 Heterotrait-monotrait ratio (HTMT).**

| Constructs | SCA | DP | RQ | OL | PA | SP | SOB | FS | EN | IM | EA | IBB |
|---|---|---|---|---|---|---|---|---|---|---|---|---|
| DP | – | | | | | | | | | | | |
| EA | 0.605 | – | | | | | | | | | | |
| EN | 0.279 | 0.496 | – | | | | | | | | | |
| FS | 0.57 | 0.544 | 0.253 | – | | | | | | | | |
| IBB | 0.524 | 0.63 | 0.478 | 0.513 | – | | | | | | | |
| IM | 0.456 | 0.521 | 0.551 | 0.526 | 0.63 | – | | | | | | |
| OL | 0.476 | 0.665 | 0.68 | 0.407 | 0.816 | 0.736 | – | | | | | |
| PA | 0.535 | 0.536 | 0.539 | 0.418 | 0.674 | 0.847 | 0.817 | – | | | | |
| RQ | 0.435 | 0.445 | 0.313 | 0.647 | 0.564 | 0.622 | 0.516 | 0.671 | – | | | |
| SCA | 0.467 | 0.586 | 0.586 | 0.462 | 0.62 | 0.67 | 0.719 | 0.713 | 0.704 | – | | |
| SOB | 0.548 | 0.689 | 0.525 | 0.521 | 0.718 | 0.686 | 0.727 | 0.743 | 0.729 | 0.843 | – | |
| SP | 0.581 | 0.779 | 0.459 | 0.326 | 0.629 | 0.481 | 0.671 | 0.589 | 0.323 | 0.547 | 0.661 | – |

Notes:
SCA, Scarcity, DP, Discounted Price, RQ, Review Quality, OL, Observational Learning, PA, Positive Affect, SP, Social Presence, SOB, Sense of Belonging, FS, Flow State, EN, Entertainment, IM, Informativeness, EA, Emotional Attachment, IBB, Impulse Buying Behavior.

independent variable explains the dependent variable, ranging from 0 to 1, and the larger it is, the greater the independent variable can explain the dependent variable. In addition, this study uses Bootstrapping to resample 5,000 samples to detect structural models.

As shown in Fig. 2, in this research model, the explanatory power of constructs, including positive affect, was 40.4% ($R^2 = 0.404$), flow state was 60.8% ($R^2 = 0.608$), the emotional attachment was 54.5% ($R^2 = 0.545$), and impulse buying behavior was 4.25% ($R^2 = 0.425$). On the other hand, the results of the path coefficient analysis are shown in Table 6. These stimulus (S) factors, including discounted price ($\beta = 0.214$, $p < 0.001$),

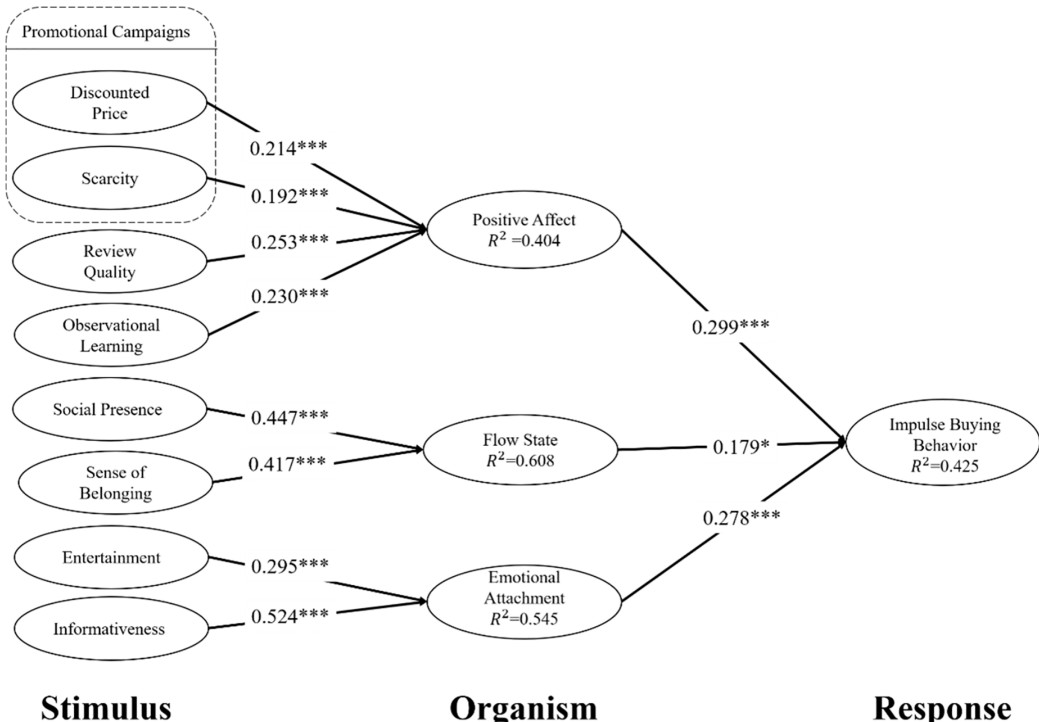

**Figure 2  Research results.** $^*p < 0.05$, $^{***}p < 0.001$.

**Table 6  Direct effect analysis.**

| Paths | | Path coefficient (β) | T-value | P-value | Result |
|---|---|---|---|---|---|
| H1a | DP → PA | 0.214 | 4.025 | 0.000 | Supported |
| H1b | SCA → PA | 0.192 | 3.967 | 0.000 | Supported |
| H2 | RQ → PA | 0.253 | 4.785 | 0.000 | Supported |
| H3 | OL → PA | 0.230 | 4.212 | 0.000 | Supported |
| H4 | SP → FS | 0.447 | 8.524 | 0.000 | Supported |
| H5 | SOB → FS | 0.417 | 8.105 | 0.000 | Supported |
| H6 | EN → EA | 0.295 | 6.422 | 0.000 | Supported |
| H7 | IM → EA | 0.524 | 11.412 | 0.000 | Supported |
| H8 | PA → IBB | 0.299 | 5.309 | 0.000 | Supported |
| H9 | FS → IBB | 0.179 | 2.526 | 0.012 | Supported |
| H10 | EA → IBB | 0.278 | 3.766 | 0.000 | Supported |

Notes:
SCA, Scarcity; DP, Discounted Price; RQ, Review Quality; OL, Observational Learning; PA, Positive Affect; SP, Social Presence; SOB, Sense of Belonging; FS, Flow State; EN, Entertainment; IM, Informativeness; EA, Emotional Attachment; IBB, Impulse Buying Behavior.

scarcity ($\beta = 0.192$, $p < 0.001$), review quality ($\beta = 0.253$, $p < 0.001$), and observed learning ($\beta = 0.230$, $p < 0.001$), had positive and significant effects on positive affect belonging to the organism (O), so H1a, H1b, H2, and H3 were supported. These stimulus (S) factors, including social presence ($\beta = 0.447$, $p < 0.001$) and sense of belonging ($\beta = 0.417$,

**Table 7 Indirect effect analysis.**

| Path | Path coefficient (β) | T-value | P-value | Result |
|---|---|---|---|---|
| DP → PA → IBB | 0.064 | 2.998 | 0.003 | Supported |
| SCA → PA → IBB | 0.057 | 2.864 | 0.004 | Supported |
| RQ → PA → IBB | 0.076 | 3.649 | 0.000 | Supported |
| OL → PA → IBB | 0.069 | 3.572 | 0.000 | Supported |
| SP → FS → IBB | 0.080 | 2.446 | 0.014 | Supported |
| SOB → FS → IBB | 0.074 | 2.318 | 0.020 | Supported |
| EN → EA → IBB | 0.082 | 3.408 | 0.001 | Supported |
| IM → EA → IBB | 0.145 | 3.480 | 0.001 | Supported |

Notes:
SCA, Scarcity; DP, Discounted Price; RQ, Review Quality; OL, Observational Learning; PA, Positive Affect; SP, Social Presence; SOB, Sense of Belonging; FS, Flow State; EN, Entertainment; IM, Informativeness; EA, Emotional Attachment; IBB, Impulse Buying Behavior.

$p < 0.001$), had a positive and significant effect on the flow state belonging to the organism (O), so H4 and H5 were supported. These stimulus (S) factors, including entertainment ($\beta = 0.295$, $p < 0.001$) and informativeness ($\beta = 0.524$, $p < 0.001$), had a positive and significant effect on emotional attachment belonging to the organism (O), so H6 and H7 were supported. Finally, the organism (O) factors, including positive affect ($\beta = 0.299$, $p < 0.001$), flow state ($\beta = 0.179$, $p < 0.01$), and emotional attachment ($\beta = 0.278$, $p < 0.001$), had a positive and significant effect on impulse buying behavior belonging to response (R), so H8, H9, and H10 were supported.

### Indirect effect analysis

Following this, the results of the indirect effect analysis are outlined. In this research, bootstrapping was utilized to evaluate the indirect effect. As illustrated in Table 7, all indirect effect paths received support. The results indicate that discounted price ($\beta = 0.064$, $t = 4.23$, $p < 0.001$), scarcity ($\beta = 0.057$, $t = 3.89$, $p < 0.001$), review quality ($\beta = 0.076$, $t = 5.12$, $p < 0.001$) and observational learning ($\beta = 0.069$, $t = 4.67$, $p < 0.001$) impact impulse buying *via* positive affect. Moreover, social interaction pathways show considerable coefficients. Social presence ($\beta = 0.080$, $t = 6.34$, $p < 0.001$) and sense of belonging ($\beta = 0.075$, $t = 5.98$, $p < 0.001$) impact impulse buying *via* flow state. Finally, content characteristics present the strongest indirect effects. Entertainment ($\beta = 0.082$, $t = 6.78$, $p < 0.001$) and informativeness ($\beta = 0.146$, $t = 8.92$, $p < 0.001$) impact impulse buying *via* emotional attachment.

## DISCUSSIONS

### Comparison of research findings

The research aims to apply the S-O-R model to investigate the factors that influence the impulsive purchase behavior of influencers on S-commerce websites. Furthermore, the sources of the stimulus were categorized into (1) posts, (2) live streaming or videos, and (3) influencers.

First, the factor about posts and message boards/chat rooms.

The present research findings show that discounted prices significantly impact the positive affect. These results are consistent with research by *Lee & Chen-Yu (2018)*, who investigated the mediating role of affective responses to price reductions on consumer perceptions in online retail contexts. Their study utilized a between-subjects experimental design featuring four different discount levels, using denim products as the primary stimulus in a simulated e-commerce environment. By systematically manipulating pricing variables across artificial web-based retail platforms, the researchers established that emotional reactions to discounted pricing were a crucial mediating factor connecting promotional tactics to consumer evaluation processes, including value perception, quality judgments, and savings awareness. Significantly, their findings indicated that when affective responses to price reductions operated as intermediary factors, the emotional stimulation created by discount exposure improved consumers' quality perceptions of the presented products.

Additionally, scarcity was determined to influence positive affect positively. This result can be contrasted with a previous study by *Yu et al. (2024)*. Their research investigated the associations between perceived scarcity, self-efficacy, self-control, and delayed gratification. They administered a survey to Chinese college students. The findings demonstrated that perceived scarcity was negatively associated with individual self-efficacy, self-control, and delayed gratification. To a certain degree, their findings elucidated how perceived scarcity impedes gratification from motivational and cognitive perspectives and supported additional research on addressing perceived scarcity's psychological and behavioral effects.

Furthermore, the current research analysis revealed that scarcity perceptions significantly enhance positive affective states among consumers.

Moreover, review quality and observational learning significantly affected positive affect, which is different from the findings of *Xu (2020)*. They found that review quality had no significant effect on positive affect. Because too many reviews lead to information overload, consumers have a negative affect. However, this study suggests that consumers can select reviews more relevant to their needs to read in the message boards of current S-commerce websites, thereby generating a positive affect.

Second, there are factors regarding live streaming and videos. Influencers can use live streaming or video on S-commerce websites to unbox or place advertorials. Social presence was found to influence the flow state significantly. The current study results align with the research by *Wei et al. (2024)*, who investigated the associations between social presence, self-confidence in academic capabilities, and flow on learner engagement in hybrid synchronous learning environments. Using a systematic literature review and thorough examination, the researchers confirmed that social presence, academic confidence, and flow significantly and directly affected learner engagement levels. Furthermore, social presence indirectly affected learner engagement by functioning through academic confidence and flow.

Furthermore, the current study's results demonstrated that sense of belonging statistically affected flow experiences. These findings align with previous research by *Bowles & Scull (2019)*, who conducted an extensive systematic review of literature from

1990 to 2016 to operationalize school connectedness and identify four component dimensions: attending, belonging, engaging, and flow. The authors then developed a sequential, four-level theoretical framework of school connectedness based on these identified dimensions. The researchers proposed that these four constructs represent the fundamental components required for engagement and theorized that flow experiences emerge due to students' psychological attachment to their educational institution. Hence, based on the above comparison, the current study indicates that with the help of advanced technology, when consumers watch live streaming or videos, they feel like they interact face-to-face with influencers, creating a social presence. In addition, they can instantly leave messages or interact with others in the chat room to increase their sense of belonging. Consumers have a social presence and a sense of belonging and are easily attracted by the current situation and atmosphere, resulting in impulse buying behaviors.

Finally, the factors regarding influencers. This study found that the entertainment and informativeness of influencers had a significant impact on emotional attachment. These findings are somewhat similar to research conducted by *Lee (2025)*. Their study aimed to examine the impact of augmented reality (AR) technology on consumer perceptions of food-related promotional materials. Participants were presented with AR-enhanced promotional content using an experimental approach and then completed self-assessment questionnaires. The results indicated that informational and entertainment values positively influenced consumer attitudes toward AR-based advertisements. Immersive experiences functioned as a full mediator in the relationship between informational content and attitudinal responses, while entertainment effects were only partially mediated through immersive mechanisms. Besides *Lee (2025)*, previous studies also indicated a positive impact of entertainment and informativeness on flow (*Ki et al., 2020*; *Sánchez-Fernández & Jiménez-Castillo, 2021*). Because with the advent of the self-media era, the number of influencers have risen abruptly. Their creations and behaviors will be followed and imitated by fans.

In addition, the research results show that positive affect significantly influences impulse buying behavior. This finding is similar to previous research by *Ibrahim, Ariyanti & Iskamto (2025)*. Their study examined purchasing behaviors within TikTok Shop's Indonesian marketplace. The research's main goal was to evaluate the statistical significance of sales promotional activities and hedonic shopping motivations on impulse purchasing behaviors among fashion product consumers in the TikTok Shop digital platform. The empirical results showed that promotional tactics and hedonic motivational elements significantly impacted consumers' impulse buying behaviors.

Furthermore, the present study showed that flow substantially influenced impulse buying behavior. This finding can be contrasted with previous research by *Husada, Prawiyadi & Andreani (2024)*. Their investigation examined the effect of hedonic browsing behaviors and flow experiences on Instagram concerning food and beverage content on spontaneous online purchasing decisions. The study utilized a survey approach to analyze these relationships. The empirical results demonstrated that flow experiences produced a statistically significant positive impact on online impulsive purchasing behavior. Likewise,

hedonic browsing exhibited a statistically significant positive effect on spontaneous online buying decisions.

In addition, this study found that consumers' emotional attachment to influencers can lead to impulse buying behavior. The present study findings can be contrasted with previous research by *Akbar et al. (2020)*. Their research analyzed the role of impulsive purchasing behavior within consumer research models. The study concentrated on retail consumer groups, and the statistical evaluations revealed that emotional brand attachment negatively impacted impulsive buying behaviors while concurrently showing a statistically significant positive association with post-purchase cognitive dissonance.

Finally, in terms of this study's mediating results, the current study used positive affect, flow experience, and emotional attachment as mediators. According to the results of the current study, flow experience, and emotional attachment were significant mediators between the indirect impacts of the independent constructs with the impulse buying behavior. The current result of employing positive affect as a mediating variable can be compared to an earlier study by *Li & Peng (2021)*. Their study employed the SOR framework, integrating flow theory and attachment theory, to examine how live streamer characteristics and live streaming environmental factors affect users' gift-giving intentions through emotional attachment and flow experiences. The empirical findings validated the mediating roles of flow experiences and emotional attachment. Therefore, the results showed that live streamer attributes can generate users' emotional attachment and flow responses toward the live streamer, thus increasing users' intentions to give gifts.

The current study's findings also supported the significant mediating role of flow experience. This finding can be compared to earlier research by *Tu, Huang & Tu (2025)*. Their study employed a cross-sectional survey approach with nonprofit sector employees to examine the impacts of social media engagement and mindfulness practices on psychological distress levels. The research's main goal was to evaluate whether positive and negative emotional states mediate these relationships. The results verified that positive affect functioned as a significant mediator in the indirect relationships between social media engagement and mindfulness practices and psychological distress outcomes.

## Theoretical implications

Next, the theoretical implications of this study are explained. First, earlier studies have analyzed the connection between S-commerce websites, live streaming, or influencer marketing and impulse buying behavior. As an illustration, *Lee & Chen (2021)* investigated the mediating effect of price discount affect, characterized as emotional activation generated by price reductions, within two different relational frameworks: initially, between price discounts and consumer perceptions covering perceived savings, quality, and value; and subsequently, between perceived value and purchase intentions in the online apparel retail setting.

However, these issues are related. Therefore, this study simultaneously examines the impact of S-commerce websites, live streaming, and influencer marketing on impulse buying behavior. This combined methodology tackles a significant theoretical deficiency by showing how numerous digital touchpoints collaborate synergistically to affect

consumer behavior. The investigation strengthens the S-O-R theory by supplying empirical validation that stimulus complexity in digital settings demands multi-dimensional interpretation rather than single-factor evaluation. Moreover, this unified framework enriches the theoretical comprehension of omnichannel consumer experiences, demonstrating how diverse digital stimuli operate together to produce more impactful psychological responses than separate channels functioning independently.

Second, *Hsu (2020)* argues that consumers watching videos from YouTube accounts can generate a sense of belonging and influence impulse buying behavior through a flow state. This study extends *Hsu*'s *(2020)* viewpoint to explore live streaming and videos on S-commerce websites. The results show that consumers who watch live streaming or videos on S-commerce websites also develop a sense of belonging and significantly affect the flow state, influencing impulse buying behavior. The theoretical significance resides in confirming the cross-platform applicability of psychological mechanisms underlying consumer engagement. This discovery reinforces the theoretical basis of flow theory in digital commerce environments and offers empirical validation for the universality of belonging-flow pathways across various social media platforms. Furthermore, the research adds to social identity theory by illustrating how digital communities cultivate psychological attachment that surpasses conventional brand-consumer relationships.

Finally, the results show that entertainment and information significantly affect consumers' emotional attachment to influencers, affecting impulse buying behavior. These results are the same as those found by *Ki et al. (2020)*. This means that although the research area is different, entertainment and informativeness are important factors affecting consumers' emotional attachment to influencers, affecting impulse buying behavior. This cross-cultural confirmation enhances the theoretical strength of para-social relationship theory in digital marketing environments, offering evidence for the universal relevance of entertainment and informativeness as core drivers of emotional connections between consumers and digital personalities. The uniformity across various geographical settings adds to the generalizability of attachment theory in modern digital commerce contexts.

Furthermore, this investigation provides several additional theoretical contributions. The research expands the conventional S-O-R model by integrating multiple organism variables concurrently, showing that positive affect (*Tu, Huang & Tu, 2025*), flow state (*Husada, Prawiyadi & Andreani, 2024*), and emotional attachment (*Pang & Zhang, 2024*) function as parallel mediating processes rather than competing routes. This multi-mediation methodology enhances our theoretical comprehension of how environmental stimuli convert into behavioral responses through intricate psychological mechanisms. The study also adds to impulse buying theory by recognizing specific digital precursors distinctive to S-commerce settings, such as observational learning and social presence, thus broadening the theoretical scope of impulsive consumer behavior beyond conventional retail environments (*Ibrahim, Ariyanti & Iskamto, 2025*). Lastly, the research progresses influencer marketing (*Choi, Wu & Lee, 2025*) theory by supplying empirical validation for the varying effects of different influencer-related stimuli on separate psychological states, providing a more sophisticated theoretical framework for

understanding how various aspects of influencer content generate specific consumer reactions.

## Practical implications

According to the research results, this study has some practical implications. First, the findings show that observational learning significantly influences positive affect, influencing impulse buying behavior. In addition, *Lu et al. (2021)* mentioned that observational learning observes the number of likes, shares, and reviews and the sales volume of products. In other words, consumers observe that sales volume also affects their positive affect and leads to impulsive buying behavior. However, most S-commerce websites do not offer sales volume to consumers. It is suggested that influencers can provide relevant data during live streaming. Alternatively, S-commerce websites can design such functions in the future. It is believed that it will be beneficial to product sales.

Beyond basic sales volume displays, platform developers should create advanced social proof systems incorporating real-time purchase alerts, peer comparison features showing "similar customers also purchased," and dynamic availability indicators that refresh based on current stock levels. These elements should be tactically placed during high engagement periods identified through user behavior analysis.

For influencer marketing strategy enhancement, organizations should establish multi-level partnership structures that utilize the varying impact of stimulus channels (*Choi, Wu & Lee, 2025*). High-entertainment influencers should be assigned to emotional connection campaigns targeting lifestyle and aspirational merchandise, while high-informativeness influencers should concentrate on practical products requiring logical validation. Marketing professionals should execute flexible content approaches where promotional activities (reduced prices, limited availability) are coordinated with influencer content schedules to maximize positive emotion enhancement during live-streaming events.

Platform designers should construct engaging social presence (*Mahmood, Jusoh & Nor, 2024*) capabilities, including virtual collaborative shopping experiences, instant peer communication during live broadcasts, and competitive community activities that strengthen a sense of community. These capabilities should integrate artificial intelligence (AI)-powered customization that adjusts social interaction levels based on individual user flow state activators, establishing optimized psychological routes to purchasing decisions.

As mentioned earlier, entertainment and informativeness significantly affect emotional attachment and lead to impulse buying behaviors. Additionally, *Casaló, Flavián & Ibáñez-Sánchez (2020)* and (*Martínez-López et al., 2020*) found that the originality and uniqueness of influencers' creations improve the willingness of fans to interact with them, follow their suggestions, and recommend their work to others. Companies that place product or brand information into influencers' creations will be better than traditional advertising. Organizations should develop thorough influencer assessment systems that evaluate audience size and emotional connection creation potential through indicators such as comment sentiment evaluation, recurring engagement frequencies, and para-social bond strength measurements. Strategic collaborations should emphasize influencers who show

reliable capability to produce flow states through content engagement and social presence stimulation. Influencer marketing will be one of the most important marketing strategies in the future.

## Research limitations and future research directions

This study tried its best, but there are still some imperfections. For example, the respondents were mostly teenagers, students, and women, which may have biased the results. Impulse buying behavior should vary by gender or age. Also, there are many categories of influencers, such as by the number of followers or the nature of their creations (*e.g.*, cuisine and cooking). However, this study did not explore the impact of different types of influencers in detail. Finally, impulse buying behaviors may lead to returns, but little research has focused on their relationship. This demographic bias potentially restricts the generalizability of results across various age cohorts, educational levels, and cultural settings where impulsive purchasing behaviors may differ considerably. Future investigations should utilize stratified sampling across varied demographic categories and incorporate multi-group examinations to explore limiting conditions.

Second, the research fails to control for essential confounding factors, including influencer classification (micro *vs.* macro influencers), product types (hedonic *vs.* utilitarian), and participants' previous experience with S-commerce platforms. These uncontrolled elements may substantially moderate the connections between stimuli and psychological reactions, potentially exaggerating or concealing actual effects. Therefore, future researchers are advised to implement experimental frameworks controlling for influencer attributes, product categories, and user proficiency levels to strengthen causal conclusions and improve theoretical comprehension of when and how S-commerce stimuli most effectively prompt impulsive purchasing behavior.

Although this study utilizes methodologically sound cross-sectional survey approaches with suitable statistical examinations, the empirical framework lacks the complexity and refinement necessary for strong scientific inference and causal comprehension. Future research should embrace a longitudinal methodology to monitor the temporal progression of psychological state modifications leading to impulsive purchases.

Future studies should broaden this S-O-R framework to include emerging technological developments transforming S-commerce environments (*Dabija & Frau, 2024*). Incorporating generative artificial intelligence in influencer marketing creates compelling research possibilities, particularly investigating how AI-created content, virtual influencers, and customized recommendation systems alter traditional S-O-R mechanisms. The emergence of multisensory extended reality metaverse settings provides exceptional opportunities to examine S-O-R processes in immersive environments (*Lăzăroiu & Rogalsk, 2024*). Future investigations should explore how sensory enhancement in virtual shopping experiences intensifies or reduces traditional psychological mediators and whether virtual presence creates unique organism states beyond current conceptual frameworks. Furthermore, algorithmic predictive modeling and sophisticated customer behavior analytics facilitate real-time stimulus enhancement

based on individual psychological characteristics. Research should investigate how dynamic customization of promotional strategies, social proof systems, and influencer alignment affects impulse buying patterns.

Finally, incorporating online trust, perceived risk, and purchase intention elements within the S-O-R framework would offer a more thorough understanding of consumer decision-making mechanisms, particularly addressing how trust mediates the connection between environmental stimuli and psychological reactions in S-commerce environments (*Lăzăroiu et al., 2020*).

# CONCLUSION

This investigation successfully employed the S-O-R model to elucidate the sophisticated psychological dynamics underlying impulsive buying conduct in influencer-facilitated S-commerce contexts. The outcomes indicate that promotional factors such as discounted pricing, scarcity indicators, and social validation elements, including review quality and observational learning, boost positive sentiment. Moreover, interactive functionalities (social presence, belonging sense) strengthen flow conditions, while influencer qualities, including entertainment and informativeness, deepen emotional bonds. The research provides theoretical value by unifying previously dispersed literature on S-commerce, live streaming, and influencer marketing within a comprehensive behavioral structure while extending S-O-R model usage to current digital landscapes. The results guide the strategic formulation of refined social proof tools, structured influencer partnership models, and immersive platform elements that streamline psychological channels to purchase outcomes. However, the cross-sectional methodology constrains causal conclusions, and subsequent research should implement longitudinal and experimental designs to establish chronological relationships.

### Funding
The authors received no funding for this work.

### Competing Interests
The authors declare that they have no competing interests.

### Author Contributions
- Tanaporn Hongsuchon conceived and designed the experiments, performed the experiments, analyzed the data, performed the computation work, prepared figures and/or tables, authored or reviewed drafts of the article, and approved the final draft.
- Shih-Chih Chen conceived and designed the experiments, performed the experiments, analyzed the data, performed the computation work, prepared figures and/or tables, authored or reviewed drafts of the article, and approved the final draft.
- Asif Khan conceived and designed the experiments, performed the experiments, analyzed the data, performed the computation work, prepared figures and/or tables, authored or reviewed drafts of the article, and approved the final draft.

## Ethics

The following information was supplied relating to ethical approvals (*i.e.*, approving body and any reference numbers):

The Institutional Ethics Committee of the National Kaohsiung
University of Science and Technology approved the study.

## Data Availability

Raw data are available in the Supplemental Files.

## Supplemental Information

Supplemental information for this article can be found online at http://dx.doi.org/10.7717/peerj-cs.3113#supplemental-information.

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
