# Peer review of "Applying the S-O-R model to explore impulsive buying behavior driven by influencers on social commerce websites"

_PeerJ Computer Science, doi:10.7717/peerj-cs.3113_

## Round 0.1 · original submission · Major Revisions

·

Basic reporting

Clear and unambiguous, professional English used throughout:
The manuscript is written in generally comprehensible English; however, the level of expression lacks academic polish and consistency. While grammatical issues are minimal, many sentences are repetitive and vague, leading to weak academic persuasiveness. The manuscript would benefit from thorough proofreading and possibly a professional English editing service to enhance precision and academic tone.

Literature references, sufficient field background/context provided:
The literature review is insufficient to justify the model and hypotheses. Although multiple references are cited, the theoretical rationale for the selection and organization of constructs is underdeveloped. Notably, the antecedents (stimuli) in the S-O-R model (e.g., review quality, entertainment, sense of belonging) are introduced with minimal critical engagement or theoretical synthesis.
Moreover, the distinction between social commerce (s-commerce) and general e-commerce is not convincingly articulated. While the introduction mentions this distinction, it is not conceptually developed or clearly tied to the unique features of the proposed research model. Strengthen the theoretical background by critically engaging with recent and foundational literature, and clearly define how s-commerce is distinct in its mechanisms and implications for impulse buying.

Professional article structure, figures, tables. Raw data shared:
No major issues noted in this area.

Self-contained with relevant results to hypotheses:
The article includes all hypothesized relationships and corresponding results, and presents them in a logically ordered manner. However, the depth of discussion and interpretation is lacking, especially regarding the indirect effects and theoretical implications of the findings.
Additionally, while all variables are defined, there is insufficient elaboration of construct definitions (e.g., "sense of belonging", "observational learning") and a lack of discussion about how these fit together conceptually in the overall framework. Include more detailed justification for variable inclusion and clearer operational definitions to enhance the theoretical coherence of the model.

Formal results should include clear definitions of all terms and theorems, and detailed proofs:
Not directly applicable, as the paper is empirical rather than mathematical. Still, construct definitions and measurement details could be more clearly presented, particularly in the methodology section. In its current form, the theoretical clarity is insufficient to guide replication or further development.

Experimental design

Original primary research within Aims and Scope of the journal:
The manuscript represents original empirical research involving a survey-based analysis of impulse buying behavior within social commerce platforms, which aligns in principle with PeerJ’s interdisciplinary scope covering computer science and information systems. However, the contribution is marginal, as the study appears to be a repackaging of existing constructs and models (e.g., the S-O-R framework) without a novel theoretical integration or research context. The study uses a conventional model in a well-explored domain with limited innovation, and therefore does not provide strong justification for its standalone contribution.

Research question well defined, relevant & meaningful. It is stated how research fills an identified knowledge gap:
Although the authors propose to explore impulsive buying behavior via influencer marketing on S-commerce platforms, the research question lacks precision and a compelling rationale.
The authors fail to clearly articulate:
What is novel about applying the S-O-R model in this context?
Why existing e-commerce studies are insufficient to understand this phenomenon?
What specific gap in social commerce research is being addressed?

Instead, the study aggregates various known variables under a familiar model without demonstrating how this fills a meaningful void in the literature. The research problem must be more sharply framed with a deeper theoretical justification. Currently, it reads more as a confirmatory model test than an inquiry into an unresolved academic issue.

Rigorous investigation performed to a high technical & ethical standard:
he study includes an ethics statement and describes the consent process for participants. The sample size (n=404) is reasonable, and common statistical techniques (CFA, SEM with bootstrapping) are used appropriately.
However, rigor is weakened by:
Sample bias (predominantly young female students), which limits generalizability.
Lack of control for confounding variables such as influencer type, product category, or user familiarity with S-commerce.

While technically acceptable, the empirical design lacks depth and sophistication needed for strong scientific inference.

Methods described with sufficient detail & information to replicate:
The authors list previous studies from which scale items were adapted and mention using a 7-point Likert scale. They also describe translation and expert review procedures. However:
The exact wording of items is not included (even in supplementary materials), limiting replicability.
Survey administration procedures (e.g., platform, duration, recruitment method) are only vaguely described.
There is no detailed explanation of data cleaning or exclusion criteria for invalid responses.

To meet PeerJ standards, replication would require full item lists, clearer sampling procedures, and transparency in data treatment.

Validity of the findings

Meaningful replication encouraged where rationale & benefit to literature is clearly stated:
The study lacks a strong rationale for replication or contextual adaptation, and fails to highlight how it meaningfully advances the field. Furthermore, the research model appears to be an aggregation of constructs from various previous studies, without a clearly articulated conceptual logic that ties them together cohesively. The integration of multiple variables feels ad hoc, and the overall theoretical justification for the model remains underdeveloped throughout the manuscript.

All underlying data have been provided; they are robust, statistically sound, and controlled:
he authors provide complete datasets and perform confirmatory factor analysis (CFA), reliability assessments (Cronbach’s alpha, CR), and structural equation modeling (SEM) in accordance with standard practices. The sample size is adequate, and measurement properties are reported transparently. The indirect effect analysis is technically sound and appropriately conducted using bootstrapping procedures. However, the interpretation of these mediating effects is underdeveloped. While all indirect paths are reported as statistically significant, the manuscript fails to elaborate on their theoretical relevance or explain why these mediations matter. The discussion does not address how these findings align with or extend previous studies, nor does it evaluate the comparative strength or strategic implications of the indirect paths. As a result, the analysis contributes little beyond statistical confirmation and does not enhance the theoretical insight of the study.

Conclusions are well stated, linked to the original research question, and limited to supporting results:
The conclusions are aligned with the hypotheses and presented in a coherent manner. However, they are largely descriptive and confirmatory, reiterating well-established relationships without offering new theoretical or managerial insight. The indirect effect analysis, while statistically significant, is interpreted only superficially and lacks meaningful theoretical elaboration. The interpretation of findings lacks depth and theoretical reflection, reducing the impact and clarity of the study's implications.

Additional comments

Thank you for your submission. The manuscript explores an important topic—impulse buying behavior in social commerce environments—but there are several substantive concerns that must be addressed.

First, the research model appears to be an aggregation of constructs from multiple prior studies without a clearly articulated conceptual foundation. The logic connecting the variables lacks coherence, and the rationale for applying the S-O-R framework in this particular context is not convincingly explained. The paper would benefit from a stronger theoretical justification for the proposed model and a clearer distinction between social commerce and general e-commerce.

Second, while the statistical procedures (e.g., CFA, SEM, bootstrapping) are executed appropriately, the interpretation of results remains largely descriptive. In particular, the indirect effect analysis, though technically sound, is discussed only at a surface level. The manuscript does not explain the theoretical significance of the mediating effects, nor does it identify which indirect paths are most meaningful from a managerial perspective.

Third, the practical implications are rather generic and do not offer actionable insights for practitioners or platform developers. Statements such as adding features to display sales volume are too simplistic and fail to reflect the strategic complexity of influencer marketing.

Lastly, the manuscript lacks novelty. It reiterates findings already well established in the literature, without offering a clear knowledge contribution or demonstrating how this study advances the field.

For these reasons, I do not recommend the manuscript for publication in its current form. I hope the authors find these comments helpful in revising or refining their research in the future.

Reviewer 2 ·

Basic reporting

see the report.

Experimental design

see the report.

Validity of the findings

see the report.

Additional comments

‘Impulsive purchasing behavior has always been an important issue for consumer behavior research, and it accounts for 40.0% to 80.0% of all types of purchasing types (Amos et al., 2014)’ – too old data that may not reflect the current situation. As the topic is very hot, make sure you substantiate your claims each time by the most recent and relevant supporting sources to reflect the current situation. ‘Because of COVID-19, many people cannot or reduce out, and then shop through the Internet’ – refer to the past, as there is no longer a COVID-19 pandemic. ‘There has been research exploring the impact of specific information communication techniques (e.g., posting) on impulse buying behavior on S-commerce websites (e.g., Leong et al., 2018; Ming et al., 2021; Zafar et al., 2021)’, ‘Researches indicate that netizens post their creations (e.g., text or video) on social media and have large followings (Khamis et al., 2017; Lou & Yuan, 2019; Al-Shehri, 2021)’, ‘some researches have pointed out that influencers regularly publish works on social media, which makes fans highly favor the brands or products they introduce, which in turn affects online purchase intention and impulse purchase behavior (Van-Tien Dao et al., 2014; Lim et al., 2017; Trivedi, 2021)’, ‘some researches have used S-O-R models to explore the factors affecting impulse buying behavior (e.g., Ming et al., 2021; Yang et al., 2021; Zafar et al., 2020)’, ‘it is emotional responses generated when consumers interact a stimulus (Ming et al, 2021; Xu et al, 2020; Chen & Yao, 2018)’, ‘Whether researched on brick-and-mortar or online stores, positive emotions have been found to be associated with impulse buying behavior (Bellini et al., 2017; Chan et al., 2017; Chen & Yao, 2018)’, ‘Past researches have indicated that people in a flow state lose their rationality and generate impulse buying behaviors (Shahpasandi et al., 2020; Wang et al., 2021; Wu et al., 2020)’, ‘previous researches have examined the relationship between S-commerce websites, live streaming, or influencer marketing and impulse buying behavior (e.g., Chen et al., 2016; Lee & Chen, 2021; Trivedi, 2021)’ – develop and clarify the specific contribution of each cited source. Applying the S-O-R model to explore the factors that lead to impulsive buying behavior by influencers on social commerce websites can be also considered in relation to generative artificial intelligence marketing (https://doi.org/10.24136/oc.3390), algorithmic predictive modeling, and customer behavior analytics in the multisensory extended reality metaverse (https://doi.org/10.24136/oc.3190), and consumers’ decision-making process on social commerce platforms in terms of online trust, perceived risk, and purchase intentions (doi: 10.3389/fpsyg.2020.00890). ‘Social commerce (S-commerce) is a business’ – use only the acronym after the first full phrase instantiation. Include results for each cited source that should be criticially analyzed. ‘Many researches have discussed online impulse buying behavior (e.g., Xu et al, 2020)’ – develop. ‘And Chan et al. (2017) argue that online environmental stimuli are the key factors that affect impulse buying. Therefore, this study uses the S-O-R model as a framework to discuss impulse buying behavior’ – whose study? Yours or Chan et al. (2017)’s? Hypotheses do not include 2023-2025 supporting sources. Some hypotheses are based on few sources. ‘Discussions’ should focus more and clearly on comparisons with other research results, as recent and relevant as possible. ‘Conclusions’ should be a distinct section, include a detailed synthesis of your results for a broader audience and develop on limitations and further research. Why numbering the reference list? More clarifications regarding the robustness of your methodology, particularly in comparison with others, are needed.

---

## Round 0.2 · accepted · Accept

The reviewers seem satisfied with the recent changes and therefore I can recommend this article for acceptance.

·

Basic reporting

The authors have made substantial improvements in the clarity, organization, and writing quality of the manuscript. The theoretical framework is now more clearly articulated, and the structure adheres well to the expected academic format.

Experimental design

The study is grounded in a solid conceptual framework (S-O-R), and the research hypotheses are logically developed. The data collection process is clearly described, with appropriate attention to sampling procedures and data quality assurance.

Validity of the findings

The statistical analyses are competently performed. The measurement model demonstrates good reliability and validity, and the structural model results are clearly reported and well aligned with the hypotheses.

Additional comments

I appreciate the authors’ responsiveness to feedback.

Reviewer 2 ·

Basic reporting

-

Experimental design

-

Validity of the findings

-